# Barriers and facilitators to pressure ulcer prevention behaviours by older people living in their own homes and their lay carers: a qualitative study

Jennifer Roddis ![ORCID],[1] Judith Dyson ![ORCID],[2] Marjolein Woodhouse,[1] Anne Devrell,[3] Karen Oakley,[4] Fiona Cowdell ![ORCID] [5]

¹School of Health and Care Professions, University of Portsmouth, Portsmouth, UK
²C-SCHaRR, School of Health Sciences, Birmingham City University, Birmingham, UK
³Patient and Public Involvement representative, Birmingham, UK
⁴Solent NHS Trust, Southampton, UK
⁵Faculty of Health Education and Life Sciences, Birmingham City University, Birmingham, UK

**Correspondence to**
Dr Jennifer Roddis;
jenny.roddis@port.ac.uk

## ABSTRACT

**Objective** To identify barriers and facilitators to pressure ulcer prevention behaviours in community-dwelling older people and their lay carers.

**Design** Theoretically informed qualitative interviews with two-phase, deductive then inductive, thematic analysis.

**Setting** The study was conducted in one geographical region in the UK, spanning several community National Health Service Trusts.

**Participants** Community-dwelling older patients at risk of pressure ulcer development (n=10) and their lay carers (n=10).

**Results** Six themes and subthemes were identified: (1) knowledge and beliefs about consequences (nature, source, timing and taboo); (2) social and professional role and influences (who does what, conflicting advice and disagreements); (3) motivation and priorities (competing self-care needs and carer physical ability); (4) memory; (5) emotion (carer exhaustion and isolation, caregiver role conflict and patient feelings) and (6) environment (human resource shortage and equipment).

**Conclusions** There is minimal research in pressure ulcer prevention in community-dwelling older people. This study has robustly applied the theoretical domains framework to understanding barriers and facilitators to pressure ulcer prevention behaviours. Our findings will support co-design of strategies to promote preventative behaviours and are likely to be transferable to comparable healthcare systems nationally and internationally.

## INTRODUCTION

Pressure ulcers (PUs) are localised injuries to the skin and underlying tissues, usually occurring over a bony prominence, resulting from pressure or pressure combined with shear.[1] Globally, a PU prevalence of 12.8% and 11.6% has been reported among hospitalised patients[2] and nursing home residents,[3] respectively. PUs lead to an increased risk of hospital admission, and treatment is costly, placing a substantial burden on care providers.[4] PUs have a detrimental impact on the quality of life of affected individuals[5 6] and are associated with increased morbidity,

**STRENGTHS AND LIMITATIONS OF THIS STUDY**

⇒ A structured theoretical approach was adopted to data collection and analysis.
⇒ Both inductive and deductive understandings of barriers and facilitators to pressure ulcer prevention behaviours were gained, leading to a detailed analysis of influencing factors.
⇒ Service users were involved in the design, analysis and reporting of this research.
⇒ Recruitment was undertaken in a single geographical area.

mortality, pain, fear and despondency.[6 7] While international guidelines recommend PU prevention behaviours, these focus on a hospital population and there is a dearth of research surrounding transferability to community settings.[8] A Priority Setting Partnership involving 500 patients, lay carers and healthcare practitioners (HCPs) identified *understanding the impact of patient and lay carer involvement* as the second most important priority for PU prevention.[9]

Existing evidence-based guidelines offer staff-intensive PU prevention strategies including (1) repositioning at least every 4–6 hours, (2) use of appropriate devices (eg, pressure-relieving mattresses), (3) frequent skin inspection and skin care, and (4) optimal nutrition and hydration.[10 11] Although these strategies are appropriate in all settings, home care has unique challenges that limit their applicability. Nursing staff and paid carers are able to spend only limited time with patients, so engagement of patients and lay carers in PU prevention is essential. However, there is limited empirical evidence regarding adherence of patients and lay carers to PU prevention guidance. To date, interventions to enhance patient and lay carer PU prevention behaviours have focused on education,

though these have had uncertain impact on knowledge or the number of new PUs.[11] This suggests knowledge might not be the only factor for non-adherence.[12]

When developing strategies or interventions to support health behaviour change, explicit use of theory allows an understanding of behavioural determinants (barriers and facilitators), thus enabling mapping of empirically tested behaviour change techniques (BCTs).[13 14] The theoretical domains framework (TDF)[15] synthesises all published theories of behaviour or behaviour change into 11 accessible theoretical domains. These domains offer a comprehensive framework of all potential determinants of behaviour (*knowledge, skills, social/professional role and identity, beliefs about capabilities, beliefs about consequences, motivation and goals, memory attention and decision processes, environmental context and resources, social influences, emotion, and action planning*). Identifying key determinants using the TDF supports identification of the BCTs most likely to be effective.[16]

### Objective
To identify barriers and facilitators to PU prevention behaviours in community-dwelling older people and their lay carers.

## METHODS
### Design
Qualitative interviews were conducted to elicit barriers and facilitators to PU prevention behaviours. The interview schedule was informed by domains of the TDF.

### Participants
Participants were older adults and lay carers. Older adults were community dwelling in their own home, aged ≥65 years, receiving community healthcare and assessed as being at risk of developing a PU (irrespective of current and previous PU status). Lay carers were those providing unpaid care (for people fitting our criteria for older adult participation), of any kind (eg, physical, household or other practical support). Exclusion criteria included lacking capacity (patients and carers), patients who had not been assessed for PU risk and carers supporting someone who did not fulfil the patient inclusion criteria. We aimed to recruit a minimum of 10 patients and 10 carers (n=20) which we expected to be sufficient to achieve data saturation given the theoretical underpinning of the study.[17]

### Patient and public involvement
In this study, we had the input of two patient and public involvement colleagues. One contributed to the study design and the other to design, interpretation of data, reading and commenting on results and writing up. We worked closely with the local carers centre which advised on and supported recruitment and hosted a dissemination event for the local community.

### Recruitment
Patient participants were recruited through HCPs at a community National Health Service (NHS) Trust who provided written information about the study. Several patients declined to take part as they felt that the study was not relevant to their circumstances. Carers were recruited through leaflets distributed through a range of outreach events and community settings, for example, carers centre, churches, community centres, voluntary organisations and supermarkets. Snowballing techniques were used to enhance recruitment. Recruitment of carers began in November 2021 and of patients in March 2022, and continued until May 2023. Potential participants were provided with information about the project, including the reasons for the study. Interested individuals were contacted by the research team approximately a week later, to discuss the study and decide if they wished to participate.

### Data collection
Written consent was gained at the start of the interview. Interviews took place in a venue of the participant's choice (usually their own home). Interviews were undertaken by female, postdoctoral researchers (CS, MW or JR) who were not known to participants prior to being recruited into the research, and who introduced themselves and their backgrounds at the start of the interviews. Participants were offered a gift voucher or BACS (bank transfer) payment post-interview. Single interviews were audio-recorded and transcribed verbatim. Throughout most interviews, only the researcher and participant were present, though on occasion, the person cared for or the carer was present. One interview was undertaken jointly with the patient and their carer. Notes were taken by the researcher as prompts; these did not form part of the analysis. Transcripts were not returned to participants for comment due to the potentially onerous nature of such checking, particularly with this group for whom it could be considered unethical given the time commitments of carers in particular.

### Data analysis
Transcripts were imported into NVivo (V.12) and underwent a two-stage analysis. After deductively categorising data to the domains of the TDF, inductive thematic analysis was undertaken according to a six-step process (familiarisation, code generation, combining codes to themes, reviewing themes, determining theme significance and reporting).[18] Data analysis and collection were iterative processes and continued until saturation was achieved. Analysis was initially undertaken by JD, MW and JR and reviewed by the author team.

## RESULTS
### Characteristics of participants
Table 1 presents participant characteristics (n=20). Patient participants (women n=4, men n=6) were aged

**Table 1** Participant characteristics*

| Pseudonym | Role | Gender | Health and self-reported mobility status of patient | Self-reported health status of lay carer |
|---|---|---|---|---|
| Pt1 | Patient | Male | Lower body paralysis, wheelchair to mobilise | |
| Pt2 | Patient | Female | Multiple long-term conditions (LTCs), poor mobility, mobilises using a walking frame | |
| Pt3 | Patient | Male | Paralysis of legs, wheelchair to mobilise | |
| Pt4 | Patient | Male | Usually mobile, currently compromised mobility post-surgery | |
| Pt5 | Patient | Female | Limited mobility using aids (walking frame, stairlift, rails, mobility scooter) | |
| Pt6[1] | Patient | Male | Multiple LTCs, usually mobile with a recent period of immobility | |
| Pt7[2] | Patient | Female | Walks short distances and uses a wheelchair | |
| Pt8 | Patient | Male | Multiple LTCs, walks at home with frame and two people, otherwise wheelchair user | |
| Pt9[3] | Patient | Male | Paraplegic, wheelchair user | |
| Pt10[4] | Patient | Female | Multiple LTCs, wheelchair user | |
| Ca1 | Wife caring for husband (now deceased) | Female | | Osteoporosis |
| Ca2[3] | Wife caring for husband | Female | | Good physical health |
| Ca3 | Wife caring for husband | Female | | Good physical health |
| Ca4[2] | Husband caring for wife | Male | | Good physical health |
| Ca5[1] | Wife caring for husband | Female | | Severe arthritis |
| Ca6[4] | Husband caring for wife | Male | | Good physical health |
| Ca7 | Daughter-in-law caring for mother-in-law and father-in-law (now in care home) | Female | | Good physical health |
| Ca8 | Son caring for mother | Male | | Good physical health |
| Ca9 | Wife caring for husband | Female | | Arthritis |
| Ca10 | Wife caring for husband | Female | | Good physical health |

*Where patient and carer were related, this is indicated by superscript matching numbers next to participant pseudonym.

68–94 years (mean age 79.5). Three had a current PU, four had previously had PU(s), two had never had a PU and one currently had and had previously experienced PUs. Where patient participants had experience of a PU, five had developed these in hospital and three at home. All patient participants had limited mobility. Lay carer participants (women n=7, men n=3) supported or had supported older relatives; 80% of these participants cared for someone who had a PU. Ages ranged from 56 to 82 years (mean 75.0). Eight lay carers were caring for their partner (husband or wife), one was caring for their mother-in-law and father-in-law, and one for their mother.

## FINDINGS

Interviews took a mean of 50 min. Six themes (domains of the TDF) and 11 subthemes were identified (figure 1). Few differences were identified between the barriers and facilitators to PU prevention experienced by carers when compared with those experienced by patients.

### Theme 1: knowledge and beliefs about consequences

Participants generally knew about the strategies to prevent PUs, except for good hydration and nutrition. This knowledge was generated from multiple sources however in most cases was acquired *after* PU development. The taboo nature of the subject hindered *pre*-PU awareness.

#### Nature of knowledge

Some found the term PU unfamiliar, for example, 'it's naivety in that I thought pressure sores were bed sores, so, I thought you only got them in bed' (Pt9). The three PU prevention strategies known to participants included repositioning and risk of friction when moving, for example, Ca5 stated: "caused by people spending too much time in one position… it limits the blood supply to the skin in that area". Many were aware of vulnerable areas particularly referring to the 'bottom' with some mentioning back, hips, heels and elbows. Devices were mentioned by most and many used aids to enhance mobility such as frames and 'grab rails' (a bar attached to a wall, usually

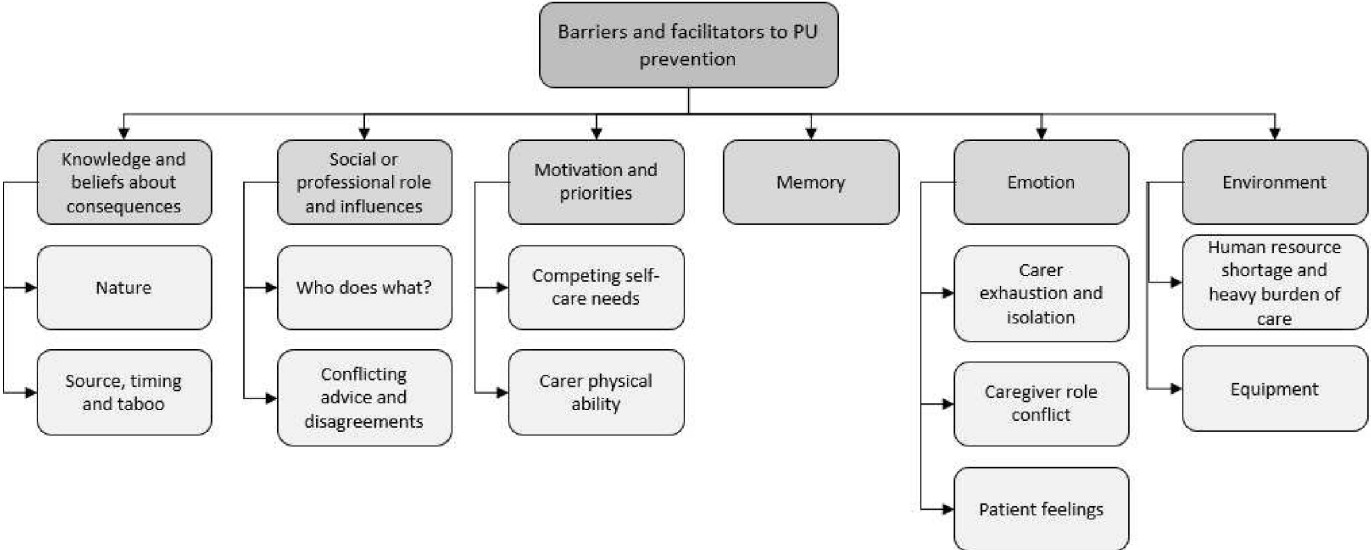

**Figure 1** Themes and subthemes. PU, pressure ulcer.

next to a bath or toilet, which the individual can hold to help them get up or down). Patient and lay carer participants reported inspecting at-risk areas of the skin and raising concerns with HCPs. Although some participants appreciated the value of good hydration and nutrition, this was rarely linked to prevention of PUs.

### Source, timing and taboo
The main source of knowledge for participants was HCPs (in hospital or at home), sometimes because they gave advice but often because the participant observed what they did and adopted these behaviours. Pt1 told us: "I've never received any information… it's really information I overhear from things like nurses… I've looked it up on the internet". Some participants had received leaflets from HCPs but found them of limited value, for example, 'if you get a wad of paper the feeling immediately is to dismiss it' (Pt1). Other knowledge sources included carers centre courses and support groups. The words of Ca1 captured the general consensus: 'there definitely needs to be more information out there', and many highlighted their lack of ability to access online information.

Participants learnt about PUs *after* they or their relative had developed one. For example, 'the ward did initially [told me about PUs], … this is what we call a pressure ulcer… [before that] I knew nothing about them' (Ca10). Information received in advance of PU development was generally not remembered. Another hindrance to PU knowledge was that they were considered unsuitable or too sensitive for discussion (a 'no-go' area), with comments such as 'it's one of those things I suppose that when you're sitting in the pub having a quiet pint with someone you don't discuss, do you?' (Pt4).

### Theme 2: social and professional role and influences
This theme included barriers relating to participants being confused as to the role of visiting practitioners,

advice which appeared to produce conflict and disagreements between practitioners or practitioners and carers.

### Who does what?
Most participants were unclear about what practitioners did, the purpose of their visit and who to approach about PUs. Online supplemental table 1 illustrates the nature and number of health and social carers with whom individuals were in contact over the previous year. In extreme cases, the volume and number of visits left participants reflecting on a sense of chaos, for example, Ca1 stated: "four people coming a day, nurses, carers, doctors …. physios somebody knocking on your door, it's hard… I've had the physio', occupation therapist… they bring the tissue viability nurse and some other ladies… two ladies, um, I think they must be physios I think". Similarly, Ca10 said: "there's an older lady who's very good and she just gets on… does whatever is needed. And then this other person will suddenly turn up and you think, why are you here?" (online supplemental table 1).

### Conflicting advice and disagreements
Participants described variations in practitioner advice and being asked to pass instructions from one to another, causing distress and awkwardness. Ca3 explained: "we're in the middle, we've got a podiatrist telling us… 'will you tell your nurses about it?' Then we've had to relay that to three or four different nurses who see [husband]". Sometimes people reported conflicts between what the professional suggested and what they, their family members or lay carers thought best. Pt3 described how his live-in carer reacted when he did not agree with the physiotherapist's advice, 'nearly took her head off… he gets a bit agitated, my carer'. On occasion, lay carers' advice took priority over practitioners': 'I take all the advice that [wife] ever gives… but in general I have faith in the NHS' (Pt6). Ca4 described an ongoing battle with district nurses regarding

pressure-relieving equipment: 'she had oedema… incredibly wet and sweaty in the plastic, which I didn't think was a very good idea… I kept taking them off and the nurses kept telling me off and putting them back on again'. Sometimes, participants were concerned disagreements would negatively impact care: 'you've got to be careful because you don't want to upset these people who are caring for [husband]' (Ca3). Similarly, patients and lay carers did not always agree on care; in these instances, the burden of the disagreement seemed more emotionally arduous for both parties. Some carers described how the person cared for found accepting care and adhering to PU prevention behaviours a challenge, 'won't take on board the seriousness of it on and read the information… [he] would say you're keeping on too much… nagging' (Ca1).

### Theme 3: motivation and priorities

Most patients and carers considered PU prevention to be important but there were two key barriers: patients having to prioritise other physical health needs and the physical restrictions of lay carers.

#### Competing self-care needs

This theme related in the most part to repositioning. Nearly all participants reported that the patient was physically unable to adopt PU advice due to physical restrictions. Most often cited were breathlessness, pain or fear of falling. 'I tend not to lie flat… I do turn on my side, but not fully' (Pt6). Pt1 said, "sometimes, I'm in so much pain in my legs I can't do anything". Inspecting vulnerable areas for deterioration was challenging due to physical restrictions. Ca1, speaking about her own vulnerability to PUs found a way around this; she said, "can't see behind… I asked my daughter to take a picture". To a lesser extent, poor health resulted in a reduced appetite, which inhibited good nutrition as a prevention behaviour.

#### Carer physical ability

While motivated to provide the best possible care for the person they cared for, some carers were restricted by, and thus had to prioritise, their own health issues: 'I have to hoist him to reposition him every time… I am very cautious how I move him now because, you know, I was 51 when I started and I'm now 73' (Ca2).

### Theme 4: memory

Five people cited memory as a barrier. Ca5 said, "he needs to be reminded [to eat and drink]" and Ca7 said, "needed to be encouraged and reminded [to change position], she would forget". Two participants suggested alarms as reminders, for example, 'setting it up on my phone, a reminder' (Ca4).

### Theme 5: emotion

The emotional burden, including exhaustion and isolation, of giving or receiving care was significant irrespective of the presence of PUs or the need for PU prevention.

Some participants spoke of conflicting carer versus partner roles and a range of other feelings.

#### Carer exhaustion and isolation

Carer exhaustion and isolation was indirectly linked with PU prevention and was a problem expressed by all carers and most patients. Carers felt frustrated, alone and in need of practical and emotional support. 'Carers… they're left in a hole on their own and all the time they're trying to get out of this hole and deal with everything… it takes a lot of emotional and, and, energy' (Ca1). Ca5 was reluctant to speak to or seek support from her grown-up children, 'I would hesitate to load too much onto them… because they care so much about [husband]'. Sometimes, carers described a mutual 'two-way' caring relationship, where support was exchanged, but this was not always the case. Spending time away from the home helped carers to see friends or attend support groups, sometimes making use of 'sitters' (a form of respite care where someone from a care company or other organisation comes to the home and spends time with the person being cared for). 'I had a sitting service… twice a week… a lifesaver… it doesn't have to be that you go out' (Ca1). Social contact for some was nearly impossible, as Ca9 told us, "if we went somewhere socially, there's not always a disabled toilet and there is no grabrail… I miss people". Emotions impacted on interactions, leading to feelings of guilt, 'there have been [times] I have lost the plot… I feel awful… I have a flare up… he says he understands, and he does' (Ca3). Lay carers sometimes found professional care visits an intrusion of privacy and closeness, 'my house was no longer my home, I lost that. And I think we lost a little bit of our closeness' (Ca2). Beyond exhaustion and isolation, carers listed many emotions, for example, fear of their partner dying before them, or of their partner being left behind should they die, guilt that their care was insufficient or inadequate, worries about finances, frustrations of trying to get additional health or social care help, the vulnerability of needing to rely on others and distress at seeing a relative in pain or unwell.

#### Caregiver role conflict

Partner carers described personal role conflict and expressed differing opinions about the extent of their carer role. For example, 'I personally didn't help [PU care], no, it came through the nurses' (Ca1); in contrast, another carer explained: "I don't particularly want anyone to look after him… intrusion of our closeness" (Ca2). Being both partner and carer was a concern. Often this was about dignity and not allowing the 'partnership' to be subsumed by a caring role. Carers dealt with this by compartmentalising. For example, Ca1 said, "we have a plan, my husband and I, it's like I'm always going to be his wife, and when I do certain things… I say to him, I've got my carer's hat on… things that aren't quite so nice… I feel I'm two different people".

## Patient feelings

Patients reported malaise, low energy and low mood adding to the challenge of PU prevention. 'The body doesn't last forever… it is frustrating… can't even do the simplest things… life isn't a bed of roses when you get older' (Pt1). Carers recognised and empathised, for example, '[it's] sad, this fiercely independent, highly intelligent man is just not able to do things anymore' (Ca5). Patients were embarrassed by younger formal carers inspecting parts of their body, feelings carers empathised with, 'I mean we laugh and joke about it… that was my way of dealing with… it's embarrassment' (Pt4) and 'I always did what I call his private bits… washed and creamed him… some of the girls are 17 and 18 years old… it's his dignity' (Ca9).

## Theme 6: environment

The key environmental barrier was human resources. Equipment was generally available but at times provided too late.

### Human resource shortage and heavy burden of care

Over half of participants spoke about the frustrations of getting help from HCPs or making appointments. '[I] want to scream… you spend half your life on the telephone trying to chase things… it takes a lot of emotional… and energy… nurses not turning up… transport doesn't turn up… that happens such a lot' (Ca1). Out-of-hours provision was a problem, 'I dread every bank holiday… if something is going to go wrong' (Ca2). Participants believed that practitioners were short of time, short of funds, overworked or stressed: 'overburdened… I know they are just overworked, spread too thin' (Ca7).

### Equipment

Equipment considered necessary by HCPs caring for patient participants was provided, including profiling beds, bed extensions, walking frames, hoists, cushions, offloading wedges, mattresses and handrails. The only concerns were lack of support to install it and delays in receiving it. For example, Pt2 remembered: "this man, he said mattress, and he plonked the tubes down. So, I said, well what do I do with them? He said, haven't you got anyone here? I said no. Oh, I'll do it, he said… the bloody fool… . he'd just left it on top [of bedding]".

## DISCUSSION

In interviews underpinned by the TDF[15] with 10 patients and 10 carers, we identified six themes representing barriers to PU prevention behaviours: (1) knowledge and beliefs about consequences, (2) social or professional role and influences, (3) motivation and priorities, (4) memory, (5) emotion and (6) environment.

Although most barriers identified were individual in nature, one significant barrier, the late acquisition of PU knowledge (*after* PU development), partly attributable to its taboo nature, suggests population-level strategies are needed. In cases of similarly taboo areas of health, such as bowel cancer, campaigns have achieved some success. A recent campaign, involving diary-style podcasts, led by a young woman with terminal bowel cancer, destigmatised 'toilet habits' and led to a 10-fold increase in online searches for bowel cancer symptoms immediately following her death.[19] For prostate cancer, there was a 10-fold increase in use of a publicly available '30-second checker' prostate screening tool after a television presenter announced his diagnosis.[20] Social media and online campaigns are not accessed by or useful to all.[21] For the populations in question, predominantly older people, this may offer only a partial solution. For example, systematic review evidence reports that 67% of older people access the internet compared with 90% of all adults.[22] In addition, recent research shows that internet campaigns are less effective for men than women.[22] We were unable to identify any research evaluating the outcome of the annual European Pressure Ulcer Advisory Panel 'Stop Pressure Ulcers' campaign designed to raise awareness of PUs. However, such campaigns may have a value in that those accessing may influence others (eg, women and younger people influencing male and older relatives) and indeed the younger people of today are our future 'at-risk' generation.

When seeking to address individual barriers, there is a need for solutions to be tailored to the individual.[23] The TDF can be used to identify the specific BCTs that are most likely to be effective.[16] This approach has been used and systematically reviewed in the case of both HCPs[24] and patients and carers.[25] These BCTs can be translated into pragmatic intervention components. For example, where knowledge is lacking, information may be the BCT of choice, where motivation is in deficit, BCTs such as goal setting, rewards and consideration of pros and cons to the behaviour in question may offer support.[16] A co-design approach involving end users[26] is most likely to result in a product that is acceptable, practicable and equitable.[27] There is published precedent,[27] and systematic review guidance,[28] for high-quality design with older people. Elements of co-design are used in the UK National Wound Care Strategy Programme, in which the Patient Experience Network has created resources by patients for patients. However, to date, focus has been on wound care rather than specifically PUs.

### Strengths and limitations

There were strengths and limitations to our study. We used a structured theoretical approach to data collection and analysis. Service users were involved in design, analysis and reporting. Most research relating to PUs considers hospital settings and healthcare practice. To the best of our knowledge, this is the first to consider community-dwelling older patients and their lay carers. We recruited from only one geographical area; although our results may be transferable, it is possible that elements of service provision may differ both nationally and internationally. While we sought to establish facilitators (as well as

barriers), these related only to the provision of equipment and were therefore not helpful in informing future solutions.

HCPs providing care to community-dwelling older people need to give timely and consistent PU prevention messages, in an accessible format to patients and lay carers. Greater clarity in roles and responsibility among care providers and across HCP and lay carer boundaries would be valuable.

As our research is, to the best of our knowledge, the first to investigate barriers and facilitators to PU prevention in community-dwelling older people, further research is needed. Now that we understand the barriers, there is an opportunity to co-design strategies to support PU prevention behaviours in older people and test these in practice.

## CONCLUSION

Little research has considered the contribution made by community-dwelling patients and their lay carers in preventing PUs. This study sought to understand the barriers and facilitators to engaging in PU prevention through robust application of the TDF. Our findings demonstrate that the key barriers relate to knowledge and beliefs about consequences, social and professional roles and influences, motivation and priorities, memory, emotion and environment. This knowledge will underpin further research required to co-design strategies to promote preventative behaviours around PU prevention for this population. Recommendations for clinical practice include ensuring that patients and their lay carers are provided with consistent, early information about PU prevention strategies and that carers are offered support. The learning from this project is likely to be transferable to comparable national and international healthcare systems.

**Acknowledgements** Dr Carol Samuel collected some of the data for this project and began the data analysis.

**Contributors** JR, MW, FC, JD, AD and KO designed the study. JR and MW contributed to data collection. JD, MW and JR contributed to data analysis. FC, JD, JR and MW drafted the manuscript. MW, JR, KO and AD critically revised the manuscript. JR is responsible for the overall content as the guarantor, had full access to the data and takes responsibility for the integrity of the data and the accuracy of the data analysis.

**Funding** This project is funded by the National Institute for Health and Care Research (NIHR) under its Research for Patient Benefit (RfPB) Programme (grant reference number: NIHR201933).

**Disclaimer** The views expressed are those of the author(s) and not necessarily those of the NIHR or the Department of Health and Social Care.

**Competing interests** None declared.

**Patient and public involvement** Patients and/or the public were involved in the design, or conduct, or reporting, or dissemination plans of this research. Refer to the Methods section for further details.

**Patient consent for publication** Not applicable.

**Ethics approval** This study involves human participants and was approved by the South West-Cornwall & Plymouth Research Ethics Committee (reference: 21/SW/0061). Participants gave informed consent to participate in the study before taking part.

**Provenance and peer review** Not commissioned; externally peer reviewed.

**Data availability statement** Data are available upon reasonable request.

**ORCID iDs**
Jennifer Roddis http://orcid.org/0000-0002-5894-5491
Judith Dyson http://orcid.org/0000-0002-0928-0438
Fiona Cowdell http://orcid.org/0000-0002-9355-8059

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
