## [Reviewer comments · BMJ Open]

ARTICLE DETAILS

TITLE (PROVISIONAL)	Barriers and facilitators to pressure ulcer prevention behaviours by older people living in their own homes and their lay carers: A qualitative study
AUTHORS	Roddis, Jennifer; Dyson, Judith; Woodhouse, Marjolein; Devrell, Anne; Oakley, Karen; Cowdell, Fiona

VERSION 1 – REVIEW

REVIEWER	Holloway, Samantha Cardiff University, Centre for Medical Education
REVIEW RETURNED	30-Dec-2023

GENERAL COMMENTS	Thank you for this interesting manuscript of a well-designed and well-executed study. The article is written clearly overall and follows the expected reporting guidelines. There are some minor comments regarding the requirement for a little more information on when the study was undertaken, and also over what period of time participants were recruited. Further information about how mobility of the patients and also health status of the carers were operationalised would be helpful to aid replication. There are also some further points to draw out in the Discussion section. Please see the comments on the PDF of the manuscript. 'The reviewer provided a marked copy with additional comments. Please contact the publisher for full details.'
---

REVIEWER	Lindhardt, Christina L. Univ Southern Denmark, Clinical Institute
REVIEW RETURNED	02-Jan-2024

GENERAL COMMENTS	I appreciate reading this article Barriers and facilitators to pressure ulcer prevention behaviours by older people and their lay carers in community settings. The article is well written and covers an area which is important within healthcare. I will be looking forward reading this article once published. I have one recommendation. The quotations are very long and can not at times be difficult to read. A suggestion could be to have a second look at them.
--

REVIEWER	Jiang, Qixia Nanjing Jinling Hospital, Department of Burns and Plastic Surgery
REVIEW RETURNED	07-Jan-2024

GENERAL COMMENTS	This is an interesting study with certain clinical significance. But it should require major modifications.
---

	1. Page 4, lines 47-48: Lay carer participants (women n=7, men n=3) supported or had supported older relatives, and all but two of these relatives had PUs. This description is not easy for readers to understand. It is recommended to modify it: Lay carer participants (women n=7, men n=3) supported or had supported older relatives, of whom, 80% of lay carers had experience taking care of the relatives with pressure ulcers. 2. Many of the expressions in Theme 1 are too colloquial, which may make it difficult for readers to understand. And it is not closely related to the theme knowledge and beliefs. Such as "Another hinderance to PU knowledge was that they were considered a "no-go" area, with comments such as "it's one of those things I suppose that when you're sitting in the pub having a quiet pint with someone you don't discuss, do you?" (Mick, patient)". Under the theme, readers would prefer to see what kind of knowledge background and beliefs of patients and carers could promote or hinder their understanding and recognition of pressure ulcers? 3.3. At the beginning of page 11, the author describes: Some carers were restricted by their own health issues, I have to hoist him to reposition him every time . . . I am very cautious how I move him now because, you know, I was 51 when I started and I'm now 73, there is a lot of difference in that age group" (Janis, carer). Which part of this description is what the carer said? Which part is the theme content extracted by the authors? Unclear and not closely related to the theme. 4.4. On page 11, regarding the emotional description of Carer exhaustion and isolation, the author seems to have paraphrased their original words, with incoherent sentences and vague semantics, requiring readers to guess their true meaning, which can be difficult for readers to understand, especially those from different cultural backgrounds. Suggest accurately expressing the original words without violating the original intention. Such as the paragraph "Sometimes carers expressed the caring relationship as "two-way", where support was exchanged, but this was not always the case. Breaks out of the house helped carers to see friends or attend support groups, sometimes making use of "sitters". "I had a sitting service . . . twice a week . . . a lifesaver. . . just for two hours . . . it doesn't have to be that you go out" (Cass, carer). Social contact for some was nearly impossible, as Diana (carer) told us, "if we went somewhere socially, there's not always a disabled toilet and there is no grabrail . . . I miss people". Emotions impacted on interactions, leading to feelings of guilt, "there have been [times] I have lost the plot . . . I feel awful . . . I have a flare up . . . he says he understands, and he does" (Nina, carer). Lay carers sometimes found professional care visits an intrusion of privacy and closeness, "my house was no longer my home, I lost that. And I think we lost a little bit of our closeness" (Janis, carer)." 5. On the page 12, about the Carer role versus partner role, there is a similar issue where the author only paraphrases the interviewee's original words and lacks sufficient refinement of the topic content. Suggest changing the theme of this section to "Carers individual Role Conflict" or "Caregiver Role Conflict". 6. On the page 12, about the section of Human resource and response, the author still paraphrased the interviewee's original words and lacks sufficient refinement of the topic content. Is it more appropriate to suggest changing the theme of this section to "Human resource shortage and heavy burden of care"? 7. On pages 12-13, regarding the obstacle factors of the equipment, the author only recounted a complaint from a patient about inadequate service provided by the installation service
--	--

	personnel of the mattress equipment. Can carers or patients correctly understand and choose the relevant equipment for use? This is a greater influencing factor for preventing PU, but the author did not describe it. Suggest supplementation. 8.The discussion section did not delve into the causes of obstacles and their potential impact on PU prevention, nor did it provide suggestions on how to eliminate or alleviate obstacles. No beneficial factors promoting PU prevention were found throughout the article, which is inconsistent with the title of this article. 9.There is no description of research limitations in this article.
--	---

VERSION 1 – AUTHOR RESPONSE

Reviewer 1's comments:	
Thank you for this interesting manuscript of a well-designed and well-executed study. The article is written clearly overall and follows the expected reporting guidelines. There are some minor comments regarding the requirement for a little more information on when the study was undertaken, and also over what period of time participants were recruited. Further information about how mobility of the patients and also health status of the carers were operationalised would be helpful to aid replication. There are also some further points to draw out in the Discussion section. Please see the comments on the PDF of the manuscript.	Thank you we appreciate your positive feedback and constructive suggestions to enhance the manuscript.
P4, line 22 - any exclusion criteria?	Exclusion criteria now included in section entitled 'Participants' (p4)
P4, line 45 - information about the period of time of the study i.e. when it commenced, period of time needed for recruitment would be good for context.	Section entitled 'Recruitment' updated to include a sentence regarding study time period (p4-5).

P4, line 47 - provided written information about the study.	
P4, line 57 - how long were participants given to decide whether they wished to take part?	Section entitled 'Recruitment' updated to include a sentence to indicate how long potential participants were given to decide whether they wished to take part (p4-5)
P5, line 19 - justification for not seeking member validation should be provided as should seeking feedback from the PPI representative.	'Data collection' section updated to justify not seeking member validation. On reflection, our reference to PPI was unnecessary and was a clumsy addition as we have been clear elsewhere that we had a PPI research partner throughout. We have justified this elsewhere in the document.
P5, line 21- there should be a section on ethical approval and ethical considerations.	The ethics statement is included at the end of the manuscript as per journal instructions.
P5, line 44 - 'Of those that had ever had a PU' – awkward wording, consider revising.	Update made to 'Characteristics of participants' to reflect comments around awkward phrasing.
P6, line 5 - 'Mobility status' – how was this defined / operationalised to ensure consistency?	Column title amended to self-reported.
P7, line 12 - 'Good physical health' was this assessed / self-reported.	Column title amended to self-reported.
P8, line 7 – not really needed	Redundant words removed.
P8, line 10 – same here, sentence will need revising	Redundant words removed and sentence restructured.
P14, line 40 – it would be good to refer to the Global Stop the Pressure Campaign here. What impact is this having and how might it be improved?	Discussion, paragraph 2 – sentence added regarding the EPUAP 'Stop Pressure Ulcers' campaign.
P14, line 47 – a little more information about which BCTs the authors feel may be most appropriate would enhance the discussion to give readers some tangible recommendation to consider.	Discussion paragraph 3 – we have added in example BCTs with references (p18).
P14, line 52 - 'co-design' - is this being addressed by the National Wound Care Strategy, and the Pressure Ulcer Workstream specifically? What inroads are being made to try and address this?	We have added a short section regarding the NWCSP.

P14, line 58 - suggest these are discussed under a subheading of Strengths and Limitations.	Discussion, paragraph 4 – new sub-heading of 'Strengths and Limitations' added.
P15, line 3 - There has been a study in nursing home residents. Appreciating this is a different population are there any similarities / differences to be found in the results compared to your own?	The Lavellee et al. article (2018) article details barriers and facilitators to preventing pressure ulcers in nursing home residents. The focus is on the views of staff (nursing home and community health services) rather than patients and their lay carers so there are fundamental differences (for example, whilst both consider knowledge to be a barrier, the Lavellee et al paper considers this from the perspective of staff, i.e. do staff have sufficient knowledge to act in a way that will prevent PUs, compared with our manuscript which focuses on the knowledge of members of the public). We have therefore not included this paper in our discussion.
P15, line 40 – provide some tangible recommendations for clinical practice and identify the gaps in the research that need further exploration.	Conclusion updated to include further research and clinical practice recommendations
Reviewer 2's comments:	
I appreciate reading this article Barriers and facilitators to pressure ulcer prevention behaviours by older people and their lay carers in community settings. The article is well written and covers an area which is important within healthcare. I will be looking forward reading this article once published.	Thank you, we look forward to this article being published. We are now building on this study to do further research in this area.
I have one recommendation. The quotations are very long and can not at times be difficult to read. A suggestion could be to have a second look at them.	Quotations throughout have been reviewed and changes made to pages 10, 11, 14, 15.
Reviewer 3's comments:	

This is an interesting study with certain clinical significance. But it should require major modifications.	Thank you for your valuable and detailed review.
Page 4, lines 47-48: Lay carer participants (women n=7, men n=3) supported or had supported older relatives, and all but two of these relatives had PUs. This description is not easy for readers to understand. It is recommended to modify it: Lay carer participants (women n=7, men n=3) supported or had supported older relatives, of whom, 80% of lay carers had experience taking care of the relatives with pressure ulcers.	'Characteristics of participants' updated to reflect reviewer's comments regarding understandability. We hope that the proposed alternative phrasing improves the understandability of this section even more than the suggested amendment.
Many of the expressions in Theme 1 are too colloquial, which may make it difficult for readers to understand. And it is not closely related to the theme knowledge and beliefs. Such as "Another hinderance to PU knowledge was that they were considered a "no-go" area, with comments such as "it's one of those things I suppose that when you're sitting in the pub having a quiet pint with someone you don't discuss, do you?" (Mick, patient)". Under the theme, readers would prefer to see what kind of knowledge background and beliefs of patients and carers could promote or hinder their understanding and recognition of pressure ulcers?	Theme 1 has been reviewed for colloquialisms. Unfortunately some of these are found in quotations from participants, and cannot be amended if we are to accurately report what was said by participants. Colloquialisms in the analysis are amended as follows:  • 'grab rail' – definition provided • 'no-go area' rephrased to provide clarity around what is meant, with the term 'no-go area' included in brackets only We've addressed this under the sub-theme 'Nature of knowledge'.
At the beginning of page 11, the author describes: Some carers were restricted by their own health issues, I have to hoist him to reposition him every time . . . I am very cautious how I move him now because, you know, I was 51 when I started and I'm now 73, there is a lot of difference in that age group" (Janis, carer). Which part of this description is what the carer said? Which part is the theme content extracted by the authors? Unclear and not closely related to the theme.	We've amended -, throughout the manuscript direct quotes form participants are italicised.
On page 11, regarding the emotional description of Carer exhaustion and isolation, the author seems to have paraphrased their original words,	We have amended this section to add clarity and explained some definitions for added clarity.

with incoherent sentences and vague semantics, requiring readers to guess their true meaning, which can be difficult for readers to understand, especially those from different cultural backgrounds. Suggest accurately expressing the original words without violating the original intention. Such as the paragraph “Sometimes carers expressed the caring relationship as “two-way”, where support was exchanged, but this was not always the case. Breaks out of the house helped carers to see friends or attend support groups, sometimes making use of “sitters”. “I had a sitting service . . . twice a week . . . a lifesaver. . . just for two hours . . . it doesn’t have to be that you go out” (Cass, carer). Social contact for some was nearly impossible, as Diana (carer) told us, “if we went somewhere socially, there’s not always a disabled toilet and there is no grabrail . . . I miss people”. Emotions impacted on interactions, leading to feelings of guilt, “there have been [times] I have lost the plot . . . I feel awful . . . I have a flare up . . . he says he understands, and he does” (Nina, carer). Lay carers sometimes found professional care visits an intrusion of privacy and closeness, “my house was no longer my home, I lost that. And I think we lost a little bit of our closeness” (Janis, carer).”	
On the page 12 , about the Carer role versus partner role, there is a similar issue where the author only paraphrases the interviewee's original words and lacks sufficient refinement of the topic content. Suggest changing the theme of this section to "Carers individual Role Conflict" or “Caregiver Role Conflict”.	The title of the sub-theme ‘Carer role versus partner role’ has been revised to ‘Caregiver role conflict’. Thank you for this suggestion.
On the page 12 ,about the section of Human resource and response, the author still paraphrased the interviewee's original words and lacks sufficient refinement of the topic content. Is it more appropriate to suggest changing the theme of this section to "Human resource shortage and heavy burden of care"?	The title of the sub-theme ‘Human resource and response’ has been revised to ‘Human resource shortage and heavy burden of care’, Thank you for this suggestion.
On pages 12-13, regarding the obstacle factors of the equipment, the author only recounted a complaint from a patient about inadequate service provided by the installation service personnel of the mattress equipment. Can carers or patients correctly understand and	We have clarified that HCPs advise on required equipment.

choose the relevant equipment for use? This is a greater influencing factor for preventing PU, but the author did not describe it. Suggest supplementation.	
The discussion section did not delve into the causes of obstacles and their potential impact on PU prevention, nor did it provide suggestions on how to eliminate or alleviate obstacles. No beneficial factors promoting PU prevention were found throughout the article, which is inconsistent with the title of this article.	We sought to establish barriers and facilitators and agree that most of what we found are indeed barriers. On the few occasions there were facilitators, which are clearly reported in the text – for example on page 10 “Participants generally knew about strategies to prevent PUs. . .”. We have reported how barriers may best be addressed in our next paper which is also under review with BMJ Open.
There is no description of research limitations in this article.	Limitations can be found below the abstract, and also towards the end of the discussion (now given a sub-heading of Strengths and limitations, as per reviewer 1’s comments). Apologies if this was previously unclear.

VERSION 2 – REVIEW

REVIEWER	Holloway, Samantha Cardiff University, Centre for Medical Education
REVIEW RETURNED	20-Feb-2024

GENERAL COMMENTS	Thank you for revising the manuscript.
--

REVIEWER	Jiang, Qixia Nanjing Jinling Hospital, Department of Burns and Plastic Surgery
REVIEW RETURNED	20-Feb-2024

GENERAL COMMENTS	This is an interesting study, doing well.
---

VERSION 2 – AUTHOR RESPONSE

Reviewer 3’s comments:	
This is an interesting study, doing well	Thank you we appreciate your positive feedback
Reviewer 1’s comments:	
Thank you for revising the manuscript	Thank you for reviewing the manuscript